# The Interplay between Whey Protein Fibrils with Carbon Nanotubes or Carbon Nano-Onions

**DOI:** 10.3390/ma14030608

**Published:** 2021-01-28

**Authors:** Ning Kang, Jin Hua, Lizhen Gao, Bin Zhang, Jiewen Pang

**Affiliations:** 1College of Environmental Science and Engineering, Taiyuan University of Technology, Taiyuan 030024, China; kangning0076@link.tyut.edu.cn; 2Taiyuan Customs District, Taiyuan 030006, China; jinhua@tyut.edu.cn; 3Key Laboratory of Coal Science, Technology of the Ministry of Education, Taiyuan University of Technology, Taiyuan 030024, China; 4College of Environment and Safety, Taiyuan University of Science and Technology, Taiyuan 030024, China; pangjiewen@tyust.edu.cn

**Keywords:** whey protein fibrils, carbon nanotubes, carbon nano-onions, composites, interaction

## Abstract

Whey protein isolate (WPI) fibrils were prepared using an acid hydrolysis induction process. Carbon nanotubes (CNTs) and carbon nano-onions (CNOs) were made via the catalytic chemical vapor deposition (CVD) of methane. WPI fibril–CNTs and WPI fibril–CNOs were prepared via hydrothermal synthesis at 80 °C. The composites were characterized by SEM, TEM, FTIR, XRD, Raman, and TG analyses. The interplay between WPI fibrils and CNTs and CNOs were studied. The WPI fibrils with CNTs and CNOs formed uniform gels and films. CNTs and CNOs were highly dispersed in the gels. Hydrogels of WPI fibrils with CNTs (or CNOs) could be new materials with applications in medicine or other fields. The CNTs and CNOs shortened the WPI fibrils, which might have important research value for curing fibrosis diseases such as Parkinson’s and Alzheimer’s diseases. The FTIR revealed that CNTs and CNOs both had interactions with WPI fibrils. The XRD analysis suggested that most of the CNTs were wrapped in WPI fibrils, while CNOs were partially wrapped. This helped to increase the biocompatibility and reduce the cytotoxicity of CNTs and CNOs. HR-TEM and Raman spectroscopy studies showed that the graphitization level of CNTs was higher than for CNOs. After hybridization with WPI fibrils, more defects were created in CNTs, but some original defects were dismissed in CNOs. The TG results indicated that a new phase of WPI fibril–CNTs or CNOs was formed.

## 1. Introduction

Whey protein is common and easily obtained from bovine milk. It was of practical significance to prepare whey protein isolate (WPI) fibrils. Nowadays, self-assembled amyloid fibrils based on whey components are an important research field [1,2,3]. Generally, amyloid fibrils are derived from the association with amyloidosis. For example, islet amyloid peptide is associated with diabetes and β-amyloid protein is associated with Alzheimer’s disease [4]. Protein fibrils can also be synthesized in vitro. Additionally, β-lactoglobulin (β-lg) can self-assemble fibrillar proteins [5,6]. The β-lg is a globular protein with a molecular weight of 18,400 g·mol^−1^ and a radius of about 2 nm [7]. It can induce fibril formation under prolonged heating (6–24 h) at 80 °C, and has a pH of 2 and low ionic strength [8]. The average length of the fibrils is 1–8 μm, with a diameter of about 4 nm [9]. The proteinaceous material in these fibrils is held together by intermolecular β-sheets [10]. During fibril formation, the amount of β-sheets is increased. 

Carbon nanotubes (CNTs) are hollow tubes made of multi-layer graphite sheets rotating and curling around the same axis at a certain angle [11]. Their diameters range from 0.4 (SWCNTs) to 100 nm (MWCNTs); their length can reach several microns; and they have superior mechanical properties, chemical stability, and a large specific surface area [12]. Carbon nanotubes are often used as filling materials to prepare nanocomposites to improve the mechanical behaviors of matrix materials. The biological applications of carbon nanotubes have also been widely studied, such as in biosensors, drug and vaccine delivery, tissue engineering [13], and new biomaterials [14]. However, pristine CNTs have poor solubility and potential cytotoxicity [15]. Attached biomacromolecules such as protein, DNA, and RNA can promote the dispersion of CNTs [16]. The physical interactions with biomacromolecules might change their biological activity in vivo [17]. After functionalization and modification, the CNTs can load different types of drugs for targeted purposes [18]. Biocompatible-CNT-based systems have the ability to load multiple therapeutic, targeting, and probing agents for cancer therapy. It has been proven that functionalized CNTs can cross the plasma membrane through different mechanisms, notably through endocytosis [19,20,21].

Carbon nano-onions (CNOs) comprise multiple concentric shells of fullerenes. Their cage-within-cage structures generate some unique physiochemical properties. Unlike any other carbon allotropes [22,23], CNOs are equally important as CNTs and fullerenes, which are ideal for drug delivery applications due to their ability to remain in systemic circulation for hours, increasing their chances of accessing the target site [24,25,26,27,28]. In tissue engineering, modified CNO scaffolds display tissue regeneration capability [28]. Far-red fluorescent CNOs have been developed for cellular imaging purposes [29]. Despite this immense potential, it appears that the role of this novel nano-system in the biomedical field has been overlooked for many years. The research on protein fibrils–carbon nanomaterial systems will be of great significance in treating human diseases, reducing the cytotoxicity of carbon nanomaterials, and developing new technologies. The formation of amyloid fibrils in vivo could lead to a variety of diseases, such as Alzheimer’s and Parkinson’s neurodegenerative diseases. Researchers are looking for substances that can inhibit amyloid fibrosis or destroy the amyloid fibrils [30,31]. Table 1 summarizes some of the studies on the interplay of carbon nanomaterials with amyloid fibrils [32]. Some studies have shown that carbon nanomaterials can interact with various biological proteins [33]. CNTs are covered by adsorbed biological macromolecules in the biological solution because of their high specific surface area and hydrophobic surface [34]. The adsorbed proteins gather on the surface of carbon nanomaterials to form a “protein crown” [34]. The interaction between CNTs and proteins also plays an important role in the formation of β-sheets. Ghule et al. found that multi-walled carbon nanotubes (MWCNTs) provided interaction surfaces for protein adsorption or encapsulation. This could inhibit the ability of the nonpolar surface of proteins to bind protein fibrils, therefore preventing protein from further fibrosis [35]. Jana and Sengupta [36] and Wei et al. [37] studied the self-assembly of Aβ-peptide in the presence of single-walled carbon nanotubes (SWCNTs) by using molecular dynamics (MD) simulation. The Aβ-peptide is a short amphiphilic peptide, and its aggregation is closely related to the pathogenesis of Alzheimer’s disease [38]. The strong hydrophobic effect of CNTs can help locate peptides on the surface of SWCNTs. This prevents the diffusion and inhibits fibrosis of peptides. Proteins such as insulin, lysozyme, β-lactoglobulin, and cytochrome c can pattern on graphite [39,40]. This nanopatterned graphite is capable of template-guiding the alignment of amyloid fibrils [39]. The interaction between fullerenes and protein materials has also been studied. Through ThT fluorescence measurements, Kim and Lee found that fullerene could inhibit the fibrosis of protein. Fullerene could specifically bind to the central hydrophobic motif KLVFF, thus hindering the aggregation of Aβ-peptide [41]. It was found that hydrated fullerenes could not only destroy mature amyloid fibrils but also prevent the formation of new fibrils [42]. Podolski et al. found that hydrated fullerenes could effectively block the aggregation of Aβ25–35 [43]. There are few studies on the interplay between CNOs and amyloid fibrils. CNOs are a new allotrope with low toxicity and good biocompatibility. The study of the interaction between CNOs and amyloid fibrils is desirable.

On the other hand, some carbon nanomaterials have been combined with biological macromolecules to prepare hybrid nanocomposites for tissue engineering or drug delivery because of their mechanical and electrical advantages [55,56,57]. The amyloid fibrils also have certain mechanical behaviors and amino acid surfaces, which are used to prepare nanowires [58], hydrogels [59], fibrous cell scaffolds [60,61], and solid functional organic film [62]. The proteins are attached to the surfaces of CNTs in the form of monomers or oligomers [63,64], so as to improve their water solubility and reduce their cytotoxicity. CNTs change the structural properties of protein fibrils through hybridization and recombination in order to target delivery of therapeutic drugs in vivo and destroy cancer cells [64,65]. Hendler et al. used the “co-assembly” method to form hybrid amyloid-fullerene composite fibrils [66], which are used for the preparation of color separation nanomarkers, diagnostic materials, and optoelectronic devices.

The special properties of protein fibrils and carbon nanomaterials (such as the mechanical and electromagnetic properties of carbon nanomaterials and the biological properties of protein materials) can benefit each other, and their combination will greatly broaden the application ranges of these two kinds of nanomaterials. However, there is still a long way to go to fully understand the interaction between protein fibrils and carbon nanomaterials. In this research, we studied the interaction of WPI fibrils with CNTs (or CNOs) and characterized the composites of WPI fibril–CNTs (or CNOs) by SEM, TEM, XRD, Raman, FTIR, and TG. WPI fibrils were prepared by using an acid hydrolysis induction process. WPI fibril–CNTs (or CNO) composites were made using hydrothermal synthesis.

## 2. Materials and Methods

### 2.1. WPI Fibril Formation

WPI-1 was purchased from Davisco Foods International Inc. (97.8% without lecithin, NM, USA) and WPI-2 was purchased from Hilmar ingredients (90.39% with lecithin, Hilmar, CA, USA). A stock solution (about 6 wt.%) was made by dissolving WPI in Millipore water. The pH of the solution was then adjusted to 4.75 by adding 1 M HCl, followed by centrifugation (10,000 rpm, 60 min, 4 °C) and filtration of the supernatant (FP 030/0.45 μm, Schleicher and Schuell). After filtration, the pH of the filtrated solution was set to 2 by using 6 M HCl. The protein concentration of the stock solution was determined using an UV spectrophotometer (UV-1800PC, MAPADA, Shanghai, China) and a calibration curve of known WPI concentrations at a wavelength of 278 nm. The stock solution was diluted to a protein concentration of 2 wt.% with HCl solution of pH 2. The WPI solution was then heated and stirred (about 290 rpm) for 20 h at 80 °C to form fibrils.

### 2.2. CNTs and CNOs Preparation

#### 2.2.1. Preparation of CNTs

Preparation of the La_2_NiO_4_ catalyst: La(NO_3_)_3_·6H_2_O and Ni(NO_3_)_2_·6H_2_O (molar ratio of La/Ni = 2:1) were dissolved in deionized water, then citric acid was added. The solution was heated at 80 °C for 1 h with stirring, and finally it turned into a colloidal substance. The colloidal substance was calcined in a muffle furnace (10 °C/min in air; 300 °C for 1 h, then 800 °C for 5 h).

Catalytic chemical vapor deposition (CVD) of methane to make CNTs: The fixed-bed gas–solid catalytic reactor was adopted for methane CVD to make CNTs. The La_2_NiO_4_ catalyst (0.5 g) was placed inside quartz boats in a tubular quartz reactor. Firstly, nitrogen (30 mL/min) was used to flush the reactor for 30 min, then hydrogen (10 mL/min) was used to reduce La_2_NiO_4_ at 600 °C for 1 h. Afterwards, the gas was switched to methane (60 mL/min) for catalytic CVD at 800 °C for 8 h to synthesize CNTs.

Purification of CNTs: The CNTs mixed with catalysts were purified in 0.1 M nitric acid at 80 °C with stirring for 5 h. It was filtered and washed with deionized water five times. Finally, the sample was dried at 120 °C for 6 h.

#### 2.2.2. Preparation of CNOs

Pretreatment of stainless steel mesh carrier: SS316 stainless steel meshes measuring 20 mm × 20 mm were ultrasonically cleaned for 30 min in 0.1 M HCl solution. Then, the meshes were placed in a tubular quartz reactor. Nitrogen gas carrying water vapor (90 °C water vapor) was introduced into the quartz tube. The quartz tube was heated to 300 °C for 1 h. The surface of the stainless steel was used as a catalyst carrier after such treatment.

Loading of catalyst: The above pretreated stainless steel mesh was immersed in nickel oxalate solution. Citric acid was added with stirring for 1 h. The solution was heated at 80 °C and finally turned into a colloid. The colloid and stainless steel meshes were put into a crucible and calcined in a muffle furnace (Zhonghuan, Tianjin, China) at 900 °C (10 °C/min, in air) for 3 h. Finally, the stainless steel mesh loaded with catalyst was obtained.

Catalytic CVD of methane to make CNOs [67]: A fixed-bed gas–solid reactor (Zhonghuan, Tianjin, China) was also used. The stainless steel mesh catalyst was placed in a quartz tube. Nitrogen (30 mL/min) was used to purge the reactor at room temperature for 1 h, then the temperature was raised to the reaction temperature of 900 °C and the nitrogen was switched to methane (30 mL/min) for 8 h for catalytic cracking. At last, the methane was switched back to nitrogen gas and the reactor was cooled down to room temperature. Finally, the stainless steel mesh catalyst and CNOs were taken out.

Purification of CNOs: A CNOs sample was first sieved to remove the free catalyst particles. It was then mixed with concentrated HNO_3_ and refluxed at 90 °C for 40 h. After dilution and cooling, it was centrifuged at 4000 rpm for 10 min and the acid solution was removed. The remaining CNOs were rinsed thoroughly using distilled water several times until reaching neutral pH. Finally, the purified CNOs were dried.

### 2.3. Preparation of WPI Fibril–CNTs (or CNOs)

WPI fibril–CNTs (or CNOs) were synthesized using the hydrothermal method. The CNTs (or CNOs) with concentrations of 0.05 wt.%, 0.10 wt.%, and 0.15 wt.% were mixed in deionized water and treated ultrasonically for 30 min to disperse as well as possible. The same volume of WPI fibril solution was added and mixed with magnetic stirring for 30 min. The mixture was then poured into the autoclave reactor (Hongchen, Xi’an, China) for hydrothermal reaction (80 °C, 20 h). Afterwards, the product was cooled down to room temperature and the autoclave was opened and the mixture taken out. The product was dried in an oven (60 °C) for 48 h.

### 2.4. Characterization

Scanning electron microscopy (SEM): The surface morphology and structure of the sample were analyzed using a JSM-7100F scanning electron microscope (JEOL, Tokyo, Japan). The SEM photos were clearer after being sprayed with gold for 10 min before observation using a transmission electron microscope (TEM, JEM-2010, Tokyo, Japan). The sample was diluted and ultrasonically dispersed. A droplet of the solution was put onto a carbon support film on a copper grid. After 15 s, the excess part was removed with a filter paper. Subsequently, a droplet of 2% uranyl acetate was put onto the grid and again removed after 15 s. Electron micrographs were taken using a JEOL electron microscope (JEM-2010, Tokyo, Japan) operating at 100 kV.

Fourier transform infrared spectrum (FTIR): A Fourier transform infrared spectrometer (Nicolet iS10, Thermo Fisher Scientific, Waltham, MA, USA) was used. The composite material and potassium bromide were weighed at the mass ratio of 1:100 and ground under an infrared lamp for 10 min to make them evenly mixed. After compression, the FTIR spectra were recorded. The scanning range was 400~4000 cm^−1^ and the resolution was 4 cm^−1^.

X-ray diffraction (XRD): The crystal structures of the composites were characterized using a MAXima-X XRD-7000 X-ray diffractometer (Tokyo, Japan) with the following settings: Cu Kα- ray, 40 kV, 2θ from 5° to 80°.

Raman spectroscopy: Raman spectra were determined on a HORIBA HR800 (Paris, France) with a 514 nm laser.

Thermogravimetry (TG): The thermal stability of the composites in air was characterized using a NETZSCH STA449 F3 synchronous thermal analyzer (Selb, Germany). The heating range was from 30 to700 °C and the heating rate was 10 °C/min.

## 3. Results and Discussion

### 3.1. WPI Fibrils

The WPI-1 (without lecithin) fibril solution was transparent and colorless (Figure 1(a1)). The fibrils could be observed through the birefringence of polarized sheets. The WPI-2 (with lecithin) fibril solution was brown in color (Figure 1(a2)). Due to their dark color, it was difficult to observe the fibrils via the birefringence sheets. Wang et al. reported that their whey protein concentrate (WPC, containing lecithin) fibril solution gradually changed from transparent light yellow to dark brown within 5 h (80 °C, pH 1.8). They believed that a Maillard reaction occurred, since small peptides formed by WPC hydrolysis during the formation of the fibrils [68]. In this study, the WPI solutions with or without lecithin were both used to prepare the WPI fibril solution. This is the first time anyone has proven that browning was not due to a Maillard reaction with peptides, while lecithin was the reason for the browning of WPI in the preparation of fibrils.

The TEM results for WPI-1 (protein mass fraction of 97.80%, without lecithin) and WPI-2 (protein mass fraction of 90.39%, containing lecithin) fibrils are shown in Figure 1b,c. It can be observed that fibrils were randomly distributed in the solution. The length of the WPI fibrils was about 2 μm. Mantovani et al. evaluated the effects of soybean lecithin on the formation of whey protein fibrils. During heat treatment, the presence of soybean lecithin had no significant effect on the fibril formation rate or protein secondary structure conformation [69]. The results in Figure 1c show that the fibrils prepared using WPI containing lecithin had a certain agglomeration and dark color, indicating that lecithin may adhere uniformly to WPI fibrils, making the color of the fibril solution darker. This is consistent with the previous observation that lecithin can darken the color of WPI.

### 3.2. CNTs and CNOs

Figure 2a,b show the TEM and HR-TEM images of CNTs, respectively. The diameter of CNTs was about 30 nm, with multi-layered graphite walls. The La_2_NiO_4_ catalyst was reduced by hydrogen before methane cracking. After reduction, the “-□-La-□-Ni-□-La-□-Ni-□-” ordered structures were formed on the perovskite-like catalyst surface (□: oxygen vacancy). The oxygen vacancy provided a place for methane adsorption on the surface. The cracking of methane was then found to occur on Ni sites near the oxygen vacancy. The structure of -□-La-□-Ni-□-La-□-Ni-□- inhibited the aggregation of Ni particles and ensured the existence of a high concentration of nanometal Ni catalysts on the surface. Nano-Ni was a necessary condition for the growth of CNTs [70].

Figure 2c,d show the TEM and HR-TEM images of CNOs, respectively. After purification, some carbon onion cores became hollow. The hollow cores measured approximately 100 nm in diameter. The HR-TEM images clearly showed the multi-layer graphitized structure of the CNOs. The Fe-Ni alloy was the nucleation center of the carbon nano-onion formation. Methane was first decomposed into carbon atoms on Fe-Ni. Carbon atoms penetrated into the alloy to form metal carbides. Around the metal carbide catalysts, methane was further cracked and formed a multi-layered graphitic structure [67]. From the HR-TEM images, it was observed that in CNTs, the graphitic layers are not exactly parallel to each other, indicating the existence of defects. In CNOs, some graphitic carbon shells networks were not perfectly closed, indicating the existence of more defects.

### 3.3. WPI Fibril–CNT (CNOs) Composites

In general, WPI fibril–CNT (or CNO) composites showed relatively uniform colloidal structures, as seen in Figure 3. Because of the highly hydrophobic surfaces of CNTs and CNOs, they were difficult to spontaneously disperse in water in their original forms. Protein fibrils were amphiphilic, which could effectively adsorb and bind to the graphite surfaces of carbon nanoparticles, providing the required water solubility and biocompatibility [71,72]. Since the whey protein fibrils were also amphiphilic, this helped to solve the dispersion problem related to CNTs and CNOs.

For the of WPI fibril–CNT sample (CNTs: 0.05 wt.%), as seen in Figure 3a, a few agglomerated CNT particles were observed in the colloidal. Some studies reported that whey protein could be an efficient and selective dispersant for CNTs of certain diameters. The possible active binding sites on the whey protein surfaces had a better match with certain CNTs’ curvatures [54]. It was speculated that in the composites with higher concentrations of CNTs, aggregations might occur.

With the addition of more CNTs or CNOs, the viscosity of the composites increased. After drying of the WPI fibril–carbon nanocomposite gels, WPI fibril–CNTs were less uniform but glossier than WPI fibril–CNOs (Figure 3c,f). The WPI fibril–CNOs could be ideal functional bio-film materials.

From Figure 3a,d, it can be seen that WPI fibril–carbon nanomaterials were all evenly gelled. Before the carbon nanomaterials were added, the WPI fibril solutions were not gelatinous at this protein concentration. Neither the individual CNTs nor CNOs were gelatinous in a water solution. Without a hydrothermal process, the mixtures of WPI fibrils and CNTs (WPI fibrils and CNOs) were not gels. Only when subjected to a hydrothermal process did the composites became colloidal. Some authors have reported that the amyloid fibril-based hydrogels could be altered in terms of both the physical and structural properties in the presence of CNTs [73]. This means that protein fibrils and CNTs interacted under certain conditions. The gel formation might be due to the following factors: (i) the fibrillar structure of WPI fibrils could promote gel formation; (ii) the heating and pressure during the hydrothermal process in the autoclave might help the composite gelatinate; (iii) carbon nanomaterials have negatively charged surfaces, which would interact with the positively charged protein fibrils to form gels, suggesting the possibility of film formation [32].

Figure 4a,e show the SEM images of WPI fibril–CNTs and WPI fibril–CNOs. The morphology of the dispersed CNTs and CNOs can be observed. The dispersion of WPI fibril–CNOs (Figure 4e) was better than WPI fibril–CNTs (Figure 4a), supporting the information in Figure 3. In the TEM images of WPI fibril–CNTs (Figure 4b) and WPI fibril–CNOs (Figure 4f), WPI fibrils and CNTs can be clearly observed; similarly, WPI fibrils and CNOs also existed. No obvious damage was observed in CNTs or CNOs after hybridization with WPI fibrils (Figure 4c,g). However, a significant reduction in the length of WPI fibrils in the composites can be seen in Figure 4d,h. The lengths of WPI fibrils were shortened from 2 μm to about 200 nm in both the WPI fibril–CNT and WPI fibril–CNO composites. The short fibrils formed small clusters. The possible reasons for this are as follows: (i) the destruction of the intermolecular force of the fibrils under steam pressure in the autoclave; (ii) the Brownian motion of carbon nanoparticles under pressure also might cause the WPI fibrils to break down; (iii) the β-folded fibril bundles near the turning point of WPI fibrils were distorted and destroyed [74,75]. These results indicate that under hydrothermal conditions, CNTs and CNOs might destroy WPI fibrils and inhibit further protein fibrosis. This finding might have important research value in the future in targeted therapy of organ fibrosis and in vivo protein fibrosis. By using molecule simulation, researchers reported that carbon nanotubes and fullerene prevented the secondary structure formation of amyloid-β peptide oligomers [76,77,78].

Figure 5 shows the FTIR results for the WPI fibril–carbon nanocomposites. In general, it was clear that the functional group signals on the WPI fibril–CNOs were stronger than those on the WPI fibril–CNTs, demonstrating a stronger interaction between WPI fibrils and CNOs. This might be beneficial to the dispersion of CNOs and to forming a homogeneous gel. This result was consistent with the visual observation. The stretching vibration peak of the hydroxyl group appeared at 3500 cm^−1^, and the stretching vibration peak of N–H of the amide I band appeared at about 3280 cm^−1^. The peak between 3000 and 2800 cm^−1^ was the stretching vibration of the C–H bond. The absorption band in 1400–1300 cm^−1^ could be attributed to the variable angle vibration of the C–H and C–OH vibration. The range of 1260~1000 cm^−1^ was caused by C–OH stretching vibration. In acidic aqueous solution, it was easier for CNTs and CNOs to carry hydroxyl groups on the surface [79].

In Figure 5a, the absorption peaks did not change significantly with the increase in the amount of CNTs. This suggests that the increase of CNTs did not significantly compound more WPI fibrils, implying that the interaction between WPI fibrils and CNTs was weak. In Figure 5b, however, the intensity of absorption peaks increased with the increase of CNOs, indicating that more CNOs would compound more WPI fibrils.

The characteristic peaks of FTIR spectra could be used to analyze not only the functional groups of the composites, but also the secondary structures of the proteins. It can be seen from Figure 5 that the vibration types of the amide band were as follows: stretching vibration peak of amide I band C = O (1640 cm^−1^), bending vibration of amide II band in N-H plane, and characteristic absorption peak of C–N stretching vibration (1570–1520 cm^−1^). The peak patterns of amide I bands and Ⅱ band were not affected by the side chain structure of the protein, rather only by its secondary structure. The change of the protein secondary structure was analyzed by comparing the spectra of the amide I band region [80]. The amide Ⅱ band sensitively reflected intermolecular or intramolecular hydrogen bond association.

The secondary structures of proteins were mainly in the forms of the α- helix, β- fold, β- turn, and random coils. The WPI fibrils consisted of the secondary protein structures. For the WPI fibril–CNTs, the stretching vibration peak of the amide I band did not change significantly with the increase of CNTs, which revealed that the secondary structure of the WPI fibrils are not clearly influenced by the addition of more CNTs. For WPI fibril–CNOs (Figure 5b), with additional CNO content, the stretching vibration peak of the amide I band changed more significantly, implying that CNOs had great influence on the secondary structure of the WPI fibrils. Comparing Figure 5a with Figure 5b, CNOs had stronger interactions with WPI fibrils and changed more significantly in terms of the protein secondary structure than CNTs.

Figure 6 shows the XRD patterns of WPI fibril–carbon nanocomposites. The CNTs and CNOs had a layered graphite structure and their diffraction peaks were similar. Normally, there were diffraction peaks at 2θ = 26.6° and 44.1°, corresponding to characteristic peaks of graphite at (002) and (101), respectively. In Figure 6, the composites exhibited protein diffraction peaks near the diffraction angles of 2θ = 9° and 19°. In Figure 6a, for WPI fibril–CNTs, the diffraction peaks of CNTs were very weak. The reason could be that most of the CNTs were wrapped with WPI fibrils. In the XRD of WPI fibril–CNOs (Figure 6b), the graphite layer diffraction peaks of CNOs were more obvious than those in WPI fibril–CNTs. It was assumed that some CNOs might not be completely covered with WPI fibrils.

Raman spectroscopy is a useful nondestructive tool that can be used to study the structures of carbon nanomaterials [81]. Figure 7 presents the Raman spectra of the CNTs, WPI fibril–CNTs, CNOs, and WPI fibril–CNOs. The peaks were weaker in intensity after the composite process because the concentrations of CNTs and CNOs in the composites were lower. All four samples showed two main D band (around 1310 cm^−1^) and G band (around 1560 cm^−1^) peaks in the range of 1100 to 2000 cm^−1^. The D band represents various defects in the graphitic layers, such as stacking fault disorders between adjacent graphitic layers, edge defects, and atomic defects within individual graphitic layers [82]. The G band is due to the in-plane stretching vibrations of the sp2 graphitic carbon. In highly oriented pyrolytic graphite (HOPG), with an increase in the defect in the graphitic materials, the D-band becomes intense [83]. The intensity ratio of D and G bands (*I*_D_/*I*_G_) can be used as a measure of the degree of disorder in the carbonaceous materials. In an ideal graphite nanomaterial, the D band is weaker and the G band is stronger and sharper, indicating a higher degree of long-range order and a lower impurity level [84]. From the spectra for CNTs and WPI fibril–CNTs, the D band was at 1322.73 cm^−1^ and the G band was at 1565.77 cm^−1^. It was clear that the *I*_D_/*I*_G_ in CNTs (*I*_D_/*I*_G CNTs_ = 0.49) was smaller than that in WPI fibril–CNTs (*I*_D_/*I*_G WPI fibril–CNTs_ = 0.79). This indicates the existence of more defects in the WPI fibril–CNT sample, whereas for CNOs and WPI fibril–CNOs, the D band was at 1307.64 cm^−1^ and the G band was at 1554.10 cm^−1^. The *I*_D_/*I*_G_ for CNOs (*I*_D_/*I*_G CNOs_ = 2.39) was larger than for the WPI fibril–CNOs (*I*_D_/*I*_G WPI fibril–CNOs_ = 2.14), meaning unlike in the case of CNTs, after hybridization there were fewer defects in WPI fibril–CNOs. Some defective graphite layers in CNOs might be removed. Making a comparison between CNTs and CNOs, we found that the *I*_D_/*I*_G_ in CNTs was smaller than that in CNOs, indicating the existence of more defects in CNOs than in CNTs. The HR-TEM images clearly indicated that some graphite shells in CNOs were not fully closed, supporting the existence of more defects.

Figure 8 shows the TG plots of WPI fibril–CNTs and WPI fibril–CNOs. In general, they showed quite similar trends. There were three weight loss stages in the whole temperature range. The first stage happened at temperatures of 230~320 °C (about 30 wt.%), the second weight loss occurred at temperatures of 320~520 °C (about 20 wt.%), and the third one was at temperatures of 520~650 °C (about 35 wt.% for WPI fibril–CNTs and 47 wt.% WPI fibril–CNOs). The first stage of weight loss was mainly caused by the combustion of WPI fibrils, the second stage possibly corresponded to the combustion of composites of WPI fibril–CNTs or WPI fibril–CNOs, and the third stage was associated with the combustion of CNTs or CNOs. The TG results demonstrated that there were three phases in the composites of WPI fibrils with CNTs (or CNOs). A new phase for WPI fibril–CNTs or WPI fibril–CNOs was formed after hydrothermal synthesis. The thermal stability of the new composite phase was in between the individual WPI fibrils and CNTs (or CNOs).

## 4. Conclusions

WPI fibril–CNTs and WPI fibril–CNOs were prepared via hydrothermal synthesis. WPI fibrils with CNTs or CNOs formed uniform gels and films. CNTs and CNOs shortened the WPI fibrils and formed small WPI fibrils clusters. The FTIR spectra indicated that both CNTs and CNOs interacted with WPI fibrils and further influenced the secondary structure of the WPI fibrils. The XRD analysis revealed that most CNTs were wrapped in WPI fibrils, while CNOs were partially wrapped in WPI fibrils. HR-TEM imaging and Raman spectroscopy showed that the graphitization level for CNTs was higher than for CNOs. After hybridization with WPI fibrils, more defects were created in the CNTs, however some original defects were dismissed in the CNOs. The TG results showed that a new phase of WPI fibril–CNTs or CNOs was generated.

This research found that CNTs and CNOs could degrade WPI fibrils, which might have important research potential in the treatment of diseases such as lung and liver fibrosis, Parkinson’s disease, or Alzheimer’s disease. On the other hand, CNTs and CNOs were able to be modified using WPI fibrils to increase their biocompatibility and reduce their cytotoxicity. Moreover, hydrogels composed of WPI fibrils with CNTs (or CNOs) might be new materials with applications in medicine or other fields.

## Figures and Tables

**Figure 1 materials-14-00608-f001:**
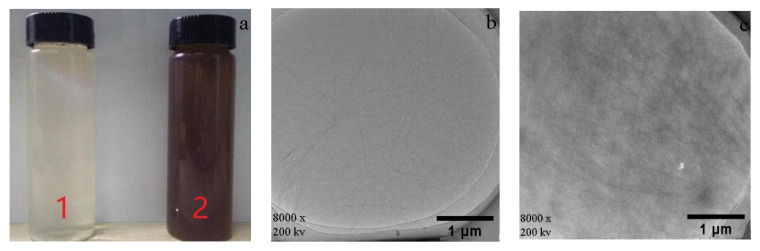
Visual appearance and TEM of fibrils prepared from WPI-1 and WPI-2 (**a1**): appearance of WPI-1 (without lecithin) fibril solution; (**a2**) appearance of WPI-2 (with lecithin) fibril solution; (**b**) TEM of WPI-1 fibrils; (**c**) TEM of WPI-2 fibrils.

**Figure 2 materials-14-00608-f002:**
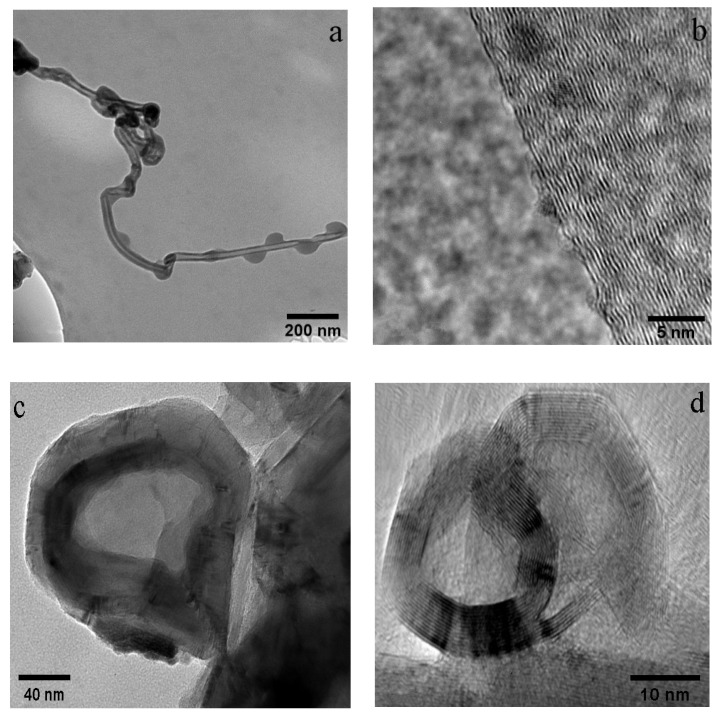
TEM and HR-TEM images of CNTs (**a**,**b**) and CNOs (**c**,**d**).

**Figure 3 materials-14-00608-f003:**
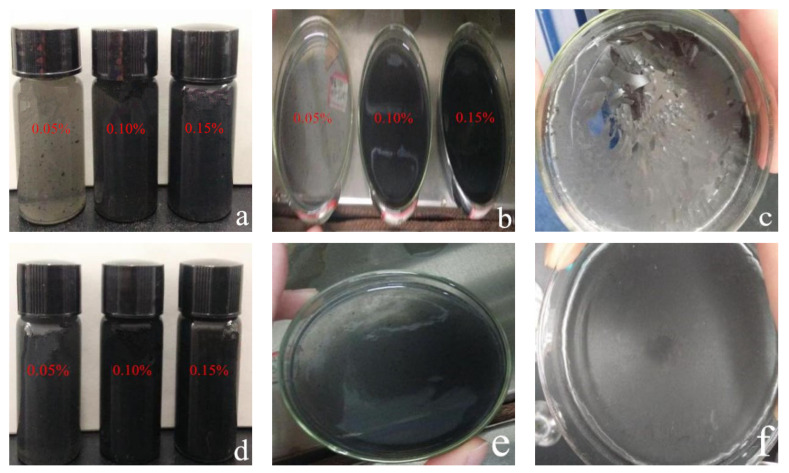
Visual appearance of WPI fibril–carbon nanocomposites: (**a**,**b**) WPI fibril–CNT composites in glass vials and dishes with different CNT concentrations; (**c**) WPI fibril–CNT (0.10 wt.% CNT) composites after drying at 60 °C for 48 h; (**d**) WPI fibril–CNO composites in glass vials with different CNO concentrations; (**e**,**f**) WPI fibril–CNO composites (0.10 wt.% CNOs) before and after drying at 60 °C for 48 h.

**Figure 4 materials-14-00608-f004:**
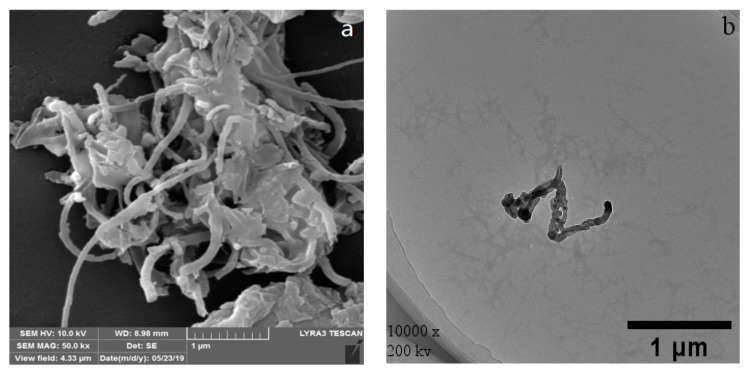
SEM and TEM images of WPI fibril–CNTs (or CNOs): (**a**) SEM image of WPI fibril–CNTs; (**b**) TEM image of WPI fibril–CNTs; (**c**) TEM image of CNTs in WPI fibril–CNTs; (**d**) TEM image of WPI fibrils in WPI fibril–CNTs; (**e**) SEM image of WPI fibril–CNOs; (**f**) TEM image of WPI fibril–CNOs; (**g**) TEM image of CNOs in WPI fibril–CNOs; (**h**) TEM image of WPI fibrils in WPI fibril–CNOs.

**Figure 5 materials-14-00608-f005:**
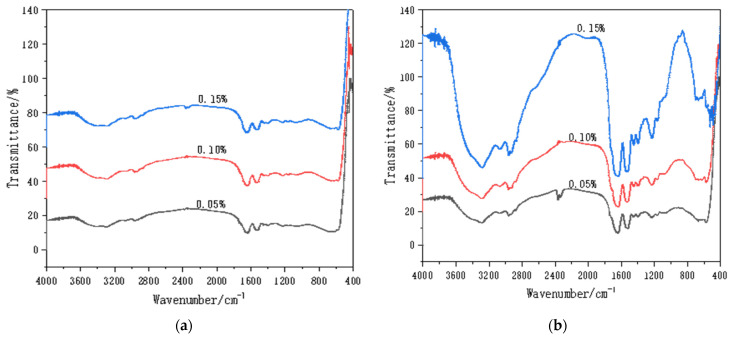
FTIR of WPI fibril–CNTs (or CNOs) (the concentrations of CNTs or CNOs were 0.05 wt.%, 0.1 wt.%, and 0.15 wt.%): (**a**) WPI fibril–CNTs; (**b**) WPI fibril–CNOs.

**Figure 6 materials-14-00608-f006:**
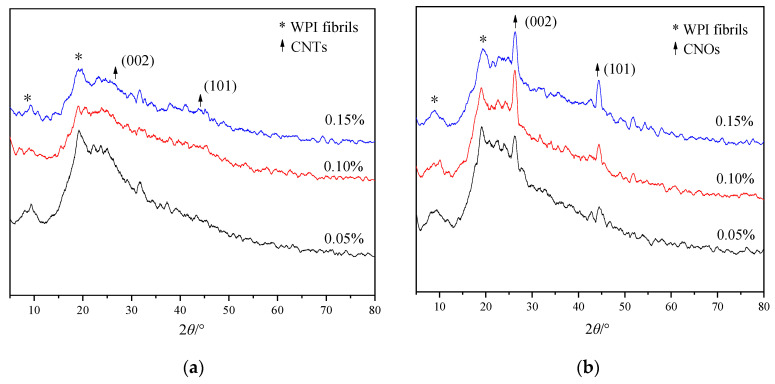
XRD patterns of WPI fibril–CNTs (or CNOs) (the concentrations of CNTs or CNOs were 0.05 wt.%, 0.1 wt.%, and 0.15 wt.%): (**a**) WPI fibril–CNTs; (**b**) WPI fibril–CNOs.

**Figure 7 materials-14-00608-f007:**
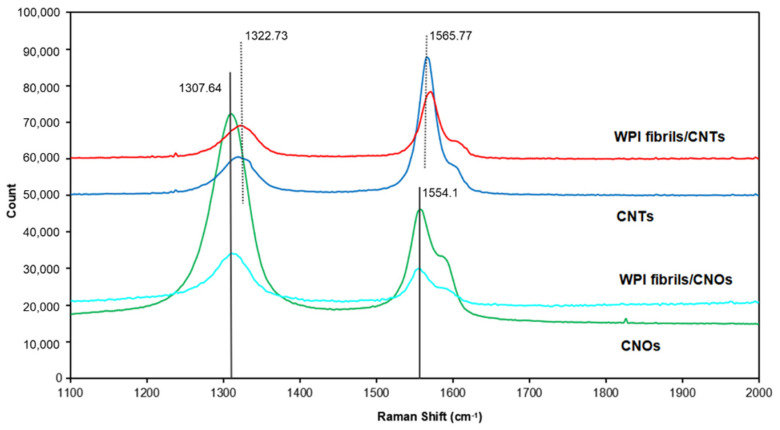
Raman spectra of WPI fibril–carbon nanocomposites and CNTs or CNOs.

**Figure 8 materials-14-00608-f008:**
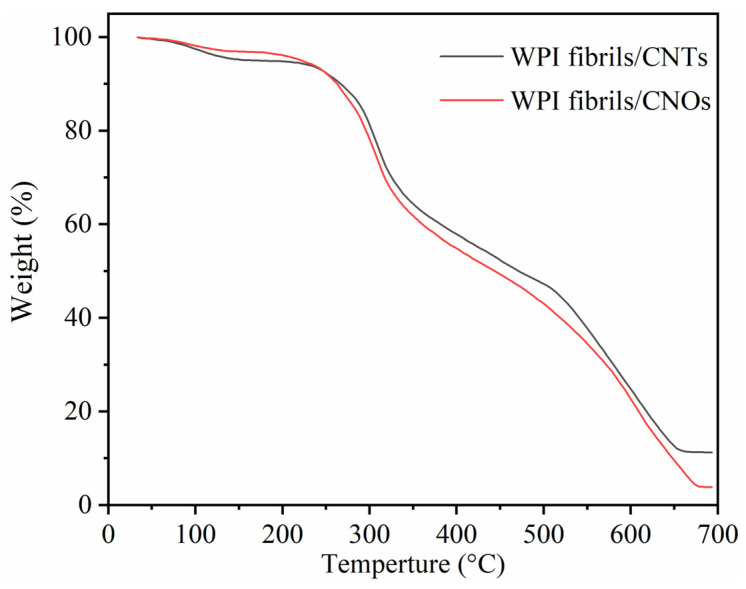
TG curves of WPI fibril–CNTs (or CNOs).

**Table 1 materials-14-00608-t001:** Brief summary of the interplay between carbon nanomaterials and amyloid fibrils.

Carbon Nanomaterials	Interplay Research on Amyloid Fibrils
Carbon nanotubes (CNTs)	Promoted protein fibrillation [44,45] Strongly inhibited the activity of Aβ peptide fibrillation [46] No reports on interplay between CNTs and WPI fibrils
Graphene/graphite	Template directed orientation of fibrillar assemblies [47,48] Graphite surfaces promoted the Aβ amyloid fibrillation [49,50] Graphene oxide inhibited the formation of fibrils [51] No reports on interplay of graphene and WPI fibrils
Fullerene	Destroyed mature amyloid fibrils [43,52] Prevented the formation of fibrils [42,53] No reports on interplay between fullerene and WPI fibrils
Carbon nano-onions (CNOs)	No reports on interplay between CNOs and amyloid fibrils No reports on interplay between CNOs and WPI fibrils Highly curved surface, lower toxicity, higher biocompatibility [54]

## Data Availability

All data, models, or code generated or used during the study are available in a repository or online in accordance with funder data retention policies.

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
