# Peer review of "The Interplay between Whey Protein Fibrils with Carbon Nanotubes or Carbon Nano-Onions"

_materials, 2021, doi:10.3390/ma14030608_

Round 1
Reviewer 1 Report
The manuscript entitled “The interplay between whey protein fibrils with carbon nanotubes or carbon nano onions” by Ning et al. seems to be of interest for the readers of the Journal. However, it is still poor written and should be much improved before acceptance for publication in the Journal.
For example, it contains numerous typographic mistakes such as: (page 1, rows 33-38) “disease[4].”, „proteins [5, 6].”, „2 nm[7].”, „strength[8].”, „4nm[9] .”, „ β-sheets.[10]” etc. Please use the coresponding typographyc spaces. In addition, (page 3 and so on, rows 111-114) „International Inc.(97.8% without lecithin),”, „centrifugation (10000rpm, 60min, 4°C)” etc.
Capital letters: (page 3, row 123) „s Nickel oxalate at the temperature of 900 oC.”
The Reference part should be corrected according to the Journal’s template and the international stadards. For example: „Ladani, L., The Potential for Metal–Carbon Nanotubes Composites as Interconnects. Journal of Electronic Materials 2019.” Page?
Besides: „Li, H.; Luo, Y.; Derreumaux, P.; Wei, G., Carbon Nanotube Inhibits the Formation of β-Sheet-Rich Oligomers of the Alzheimer's Amyloid-β(16-22) Peptide. 2011.” ???
Therefore, this manuscript should be checked and improved before resubmission for publication.
Author Response
We have revised the manuscript according to your suggestion. More details are in the attachment.

Reviewer 2 Report
RAMAN spectra need to included for ID/IG.
Lack of technical support in discussion part .
what is the advantage of this work compared to other similar works(provide the table with similar work and show the advantage of current research)
Author Response

(The authors gave the same response as above.)

Reviewer 3 Report
After careful examination, I recommend this manuscript to be rejected with the following critical comments to authors to improve the manuscript scientifically for resubmission elsewhere:
- The authors are recommended to rewrite the introduction section, especially line 44-60, page-2.
- Detailed CNO and CNT preparation is missing in section 2.2. What is the formation mechanism of CNO as well as CNT in this method of synthesis?
- How authors prepare the CNO and CNT suspension? Are they hydrophilic? How do the authors prove it?
- How authors know the CNO particles are not agglomerated as discussed in line no. 184, section 3.2?
- Most critically, how authors confirm they synthesize CNO, not the carbon soot? There are no HR-TEM results to claim it.
- CNO is structurally and physiochemically different from any other carbon allotrophs. There is no evidence of it in this manuscript. Authors are strongly recommended to have a look at recent CNO papers for their fundamental understanding. [Applied Mater. Today, 7 (2017), pp. 212-221; J. Mater. Chem. A, 1 (2013), pp. 13703-13714, etc.]
- Figure 3e-g doesn't reflect any features of CNO. Please recheck it.
- Figure 5 XRD spectra are unacceptable. Please recheck it.
- The list is not limited …......
- English must be improved throughout the manuscript. There are many awkward or grammatically incorrect expressions. Units and required spacing after that should be followed throughout the manuscript.
Author Response

(The authors gave the same response as above.)

Round 2
Reviewer 1 Report
The manuscript was much improved and can be taken into consideration for publication.
Author Response
Dear reviewer,
Thank you very much for your all kind suggestions.
best regards
Reviewer 2 Report
The new version may be published in Materials
Author Response
Thanks for all your suggestions to our manuscript.
Reviewer 3 Report
Thank you for the effort!
I think the manuscript has been modified but not improved significantly for its publication in the present form. Please modify the followings:
- carbon nano onion -> carbon nano-onion
- HRTEM -> HR-TEM
- Please use professional scale bars in Fig. 2 as it is not visible clearly. The same is applied to all other micrographs in the revised manuscript.
- Raman and HR-TEM section should contain about the graphitic structure and its associated defects as per the recommended articles Applied Mater. Today, 7 (2017), 212-221, and Mater. Chem. and Phys 174 (2016), 112-119. It has not yet discussed. Please include those well discussed Raman and microstructure papers to complete the interpretation.
- The author should correct line 437, "The ID/IG presented the disorder of the graphite ratio". English of Lines 432-444 should be checked and corrected.
- Fig. 6 XRD pattern should designate the lattice planes.
- English must be improved throughout the manuscript. There are still many awkward or grammatically incorrect expressions.
Author Response
Dear Reviewer,
Thank you for your letter and efforts to our manuscript entitled “The interplay between whey protein fibrils with carbon nanotubes or carbon nano onions” (ID: materials-1066968). We are grateful for your much helpful suggestions. We have made corrections point to point according to comments. In detail, firstly, we have identified the XRD peaks of CNO and CNT. Sencondly, we correlated the HR-TEM and Raman and made a more detailed interpretation about the Raman. Thridly, we have made a through correction in English writhing for all of the manuscript, and paid to the professional agent in Switzerland for English quality. Here we upload the certificate of the English improvement. During revision, we had a deeper understanding to our work and the manuscript quality was improved a lot ! Thanks for your kind consideration to our work.
With best regards,
Yours sincerely
Prof. Lizhen Gao
Dr. Ning Kang
Dr. Bin Zhang
